# Sustainable and Circular Hotels and the Water–Food–Energy Nexus: Integration of Agrivoltaics, Hydropower, Solar Cells, Water Reservoirs, and Green Roofs

Atour Taghipour [1,*] , Amin Padash [2], Vahid Etemadi [3], Moein Khazaei [2] and Samira Ebrahimi [3]

1 Faculty of International Business, Normandy University, 76000 Rouen, France
2 Faculty of Management and Economy, Tarbiat Modares University, Tehran 14117-13116, Iran; a.padash@modares.ac.ir (A.P.); moein.khazaei@modares.ac.ir (M.K.)
3 Faculty of Management and Economy, University of Tehran, Qom Campus, Qom 14176-43184, Iran; etemadi.vahid@ut.ac.ir (V.E.); ebrahimi.samira@ut.ac.ir (S.E.)
* Correspondence: atour.taghipour@univ-lehavre.fr

**Abstract:** The hotel industry in Iran faces critical challenges that underscore the urgent need for sustainable practices, specifically in the realms of energy, water, and food. Despite industry growth, a mere three percent of hotels fall into the five- and four-star categories, emphasizing the need for widespread adoption of sustainable practices. Focused on Ramsar in Mazandaran, the study underscores the importance of eco-friendly strategies to tackle challenges related to the food–water–energy nexus. Employing the SCOC, Fuzzy BWM, and Z-MARCOS methods, the research proposes a robust framework for evaluating hotel development strategies. The case study reveals a concentration of hotels in Khorasan Razavi, Mazandaran, and Tehran, urging prioritization of sustainable practices in these regions. Analyzing Ramsar's climate, the study suggests leveraging solar energy and implementing green roofs, emphasizing an integrated approach to achieve eco-friendly hotel construction. Furthermore, the research provides a prioritized set of strategies based on SCOC, aligning with criteria regarding the water–energy–food nexus. It emphasizes internal strengths, opportunities, and strategic technology partnerships while acknowledging external challenges such as political stability and climate change risks. The discussion introduces an Importance–Performance Analysis (IPA) to guide managerial decisions, presenting an insightful perspective for effective strategy implementation in Iran's evolving hotel industry.

**Keywords:** sustainable hotels; circular hotels; water–food–energy nexus; fuzzy; multi-attribute decision making

## 1. Introduction

### 1.1. Sustainable Hospitality

Sustainable development in the hospitality industry has become increasingly important as travelers are more conscious of their environmental and social impacts [1]. Hotels, resorts, and other hospitality businesses that adopt sustainable practices often attract a growing market of eco-conscious travelers who seek environmentally and socially responsible experiences [2,3]. It involves balancing the industry's growth and profitability with responsible and ethical considerations to minimize negative impacts on the environment, local communities, and future generations. Sustainable development in the hospitality industry encompasses several key aspects, such as environmental sustainability; social responsibility; economic viability; cultural preservation; and responsible tourism along with related certification and standards [4,5]. By incorporating these principles, the hospitality industry can contribute to the preservation of natural resources, the well-being of local communities, and the long-term success of the sector while providing guests with enriching and responsible travel experiences.

From the point of view of environmental protection, sustainable hospitality plays an important role in reducing the environmental impact of the industry. The tourism sector makes a significant contribution to carbon emissions, energy consumption, and resource depletion. By implementing environmentally friendly practices, such as energy-efficient technologies, waste reduction, and water conservation, the industry can help reduce its negative effects on the environment [6]. At the core of community engagement, sustainable hospitality encourages positive relationships with host communities. Engaging with local residents and benefiting from them through employment opportunities, supporting local businesses, and investing in community development projects can help prevent issues such as over-tourism and promote mutual understanding [7].

When it comes to economic viability in sustainable hospitality, it is not only about environmental and social responsibility. It is also about financial stability. By implementing sustainable practices, hotels and other hospitality businesses can reduce operating costs, increase revenue, and increase profitability [8,9]. Responsible tourism, as one of the main axes of sustainable hospitality, promotes responsible tourism behaviors, such as wildlife conservation, ethical cultural interactions, and environmentally friendly transportation to preserve natural and cultural assets and ensure that tourism remains a positive force [10].

Sustainable hospitality helps address global challenges, including climate change, environmental degradation, and social inequalities, while meeting the demands of increasingly conscientious travelers. Research results also show a positive outlook for the sustainable tourism market, with a projected Compound Annual Growth Rate (CAGR) of 9.54% between 2022 and 2027. The expected increase in market size to $335.93 billion is a strong growing demand for sustainable tourism (see Figure 1). Of course, this market growth depends on various factors, including large tourism companies implementing sustainable tourism practices, a shift in preference towards local and authentic experiences, and an increase in the number of travelers choosing new types of tourism (Global Sustainable Tourism Market 2023–2027, 2022).

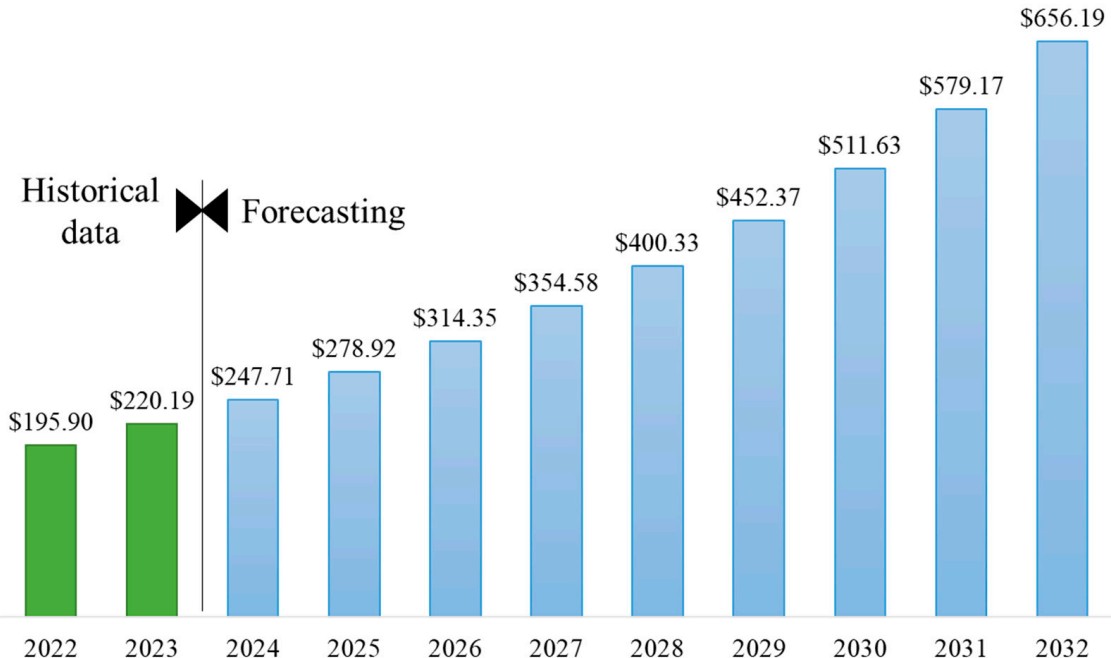

**Figure 1.** Sustainable market size outlook (billion USD) 2023–2027 (sources: Precedence Research, Global Market Insights).

### 1.2. Hospitality in Iran

Sustainable hospitality in Iran, similar to that in many other countries, faces various challenges and opportunities. A lack of awareness, growing environmental concerns,

proper management of resources, and economic factors are among these challenges. Regarding the lack of awareness, there is a need for more knowledge about, orientation to, and awareness of sustainable practices and their development and communication [11]. Environmental concerns are among the most important challenges facing the hospitality industry worldwide with the effects of global warming and loss of biodiversity. This is also true in the case of Iran and requires the use of indigenous knowledge solutions [12]. Resource management is necessary in every sector and everywhere; better management of energy and water consumption, elimination of single-use plastics, and reduction of food waste are among the things that need attention [13]. Economic factors are also among the main axes of any positive performance; budget and cost issues often create challenges for senior management in implementing sustainable practices [11].

The mentioned issues are only small but vital examples of the challenges ahead for sustainable hospitality. Iran also has significant potential for sustainable green hosting due to the diversity of natural resources and cultural heritage. Green consumer behavior is one of the important elements; there is a growing interest in green hotels among consumers in Iran. This trend is likely to continue, providing a strong market for sustainable hospitality [14]. Iran's natural resources are famous worldwide; Iran's diverse natural resources can be used for renewable and sustainable energy, which can be used to provide energy in hotels [15]. Cultural heritage is another important source of potential for sustainable hosting; Iran's rich cultural heritage and historical places attract tourists from all over the world [16]. Green and sustainable management practices are important approaches in the development of sustainable hospitality; hotels in Iran have adopted sustainable practices such as water consumption management, waste reduction, efficient management of resources, and implementation of green marketing strategies [17]. Ethical and sustainable hotels are also one of the sources of potential in sustainable hosting; currently, there are ethical and sustainable hotels in Iran that are an example for others [17,18].

### 1.3. Importance of the Water–Food–Energy Nexus in the Hospitality Industry

The water–food–energy nexus plays a pivotal role in the hospitality industry due to its profound impact on sustainability, resource management, and overall operational efficiency. Water is essential to many hospitality functions, from guest services such as swimming pools and laundry to food preparation and landscaping. It is inherently related to the production and preparation of food, which requires water for cultivation, processing, and cooking [13,19]. Energy is another critical component, as hotels require significant energy for heating, cooling, lighting, etc. The interconnectedness of water, food, and energy in the hospitality sector emphasizes the need for responsible and efficient management of resources. By understanding and optimizing this link, hotels can reduce resource consumption, reduce costs, and minimize environmental impact while simultaneously increasing guest satisfaction [20].

The water–food–energy nexus is central to sustainable development and is becoming increasingly important in the face of a rising global population, rapid urbanization, changing diets, and economic growth [21]. Connections between water, food, and energy are at the center of long-term economic and environmental development and protection. Water, energy, and food are the keys to economic input and a necessary component of economic progress. The adoption of water management policies and techniques that support the sustainable use of resources while promoting economic growth is becoming an important concern, particularly in countries where water and food scarcity are critical or problematic [22].

In the hospitality industry, this nexus becomes even more critical. The industry is a significant consumer of water, energy, and food, and its operations can have a substantial impact on the availability and quality of these resources. Therefore, understanding and effectively managing the water–food–energy nexus is essential for the industry's sustainability [23]. It ensures that the vital structures and functions of the ecosystem on which it is dependent are well protected in the face of increasing socioeconomic and climatic

stress. Moreover, sustainability in the water–food–energy nexus is essential to guarantee the responsible and equitable use of natural resources. The growing demand for these resources and the scarcity in some regions of the world make it necessary to address these challenges from an integrated and holistic perspective [21,24].

The importance of food and water as one of the main axes of sustainable development is further emphasized, according to Statista's global consumer survey data (Figure 2 (www.statista.com, Source ID: 28675, accessed on 20 December 2023), by a significant majority of survey participants in South Africa, which constituted 54% of respondents. They identified the food and water supply as the main concern for their country. Almost 40% of respondents in Mexico and 33% in Brazil share this concern. In Europe, food and water security raised concerns, with one in five in the UK, France, and Spain citing it as an important concern. Similarly, in the United States, approximately 27% of respondents indicated that this was a priority. This concern is consistent with the experience of several countries, including the United States, which have seen widespread droughts in recent summer months.

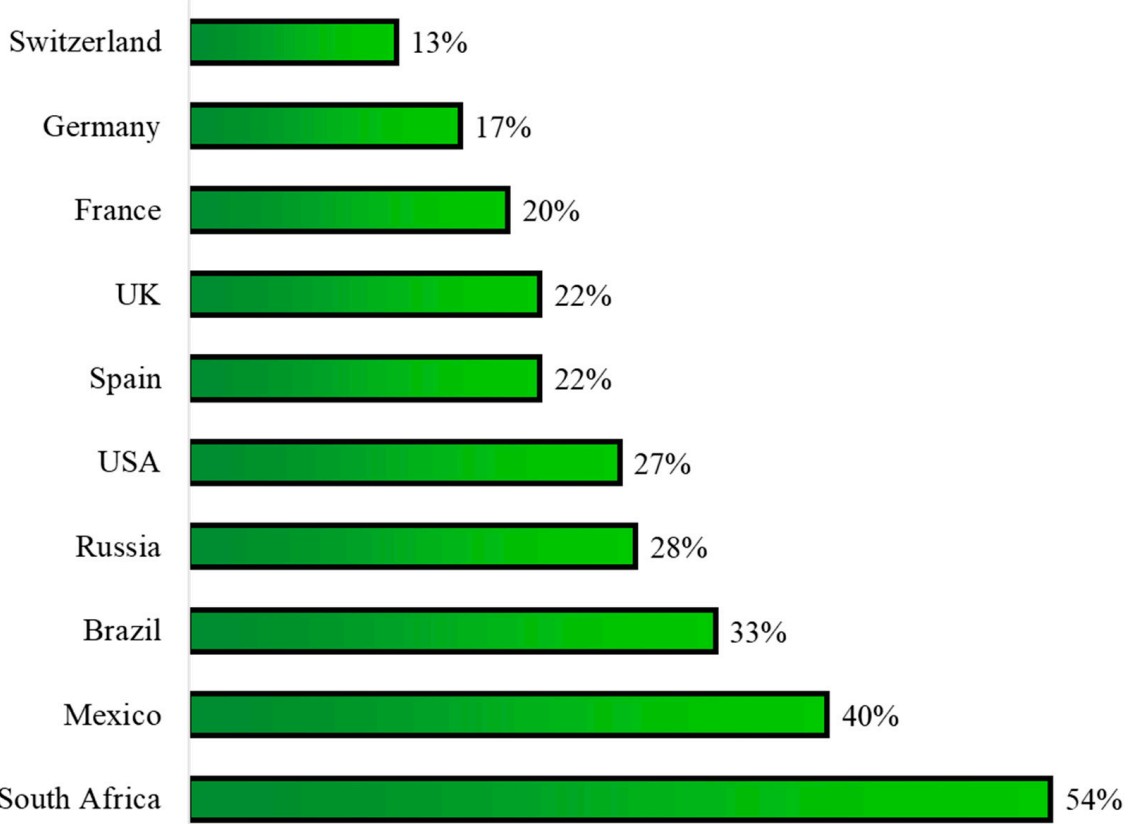

**Figure 2.** The results of the survey on the importance of the food and water supply as the main concern for the countries of the world in 2022 (source: Statista).

In conclusion, our study addresses the critical imperative of sustainable and circular hotel development, particularly within the context of the water–food–energy nexus. Through the integration of agrivoltaics, hydropower, solar cells, water reservoirs, and green roofs, we propose a comprehensive framework aimed at mitigating the challenges faced by the hospitality industry in Iran and beyond. This research endeavors to contribute to the ongoing discourse surrounding sustainability practices in the hospitality industry, particularly within the context of 4- and 5-star hotels. We aim to provide a comprehensive framework for evaluating and prioritizing sustainable strategies. While our study focuses on a specific subset of hotels, the principles and methodologies presented herein hold broader implications for sustainability efforts across various sectors. We encourage

further exploration and adaptation of these approaches to address the diverse and evolving challenges faced by businesses and organizations striving for environmental stewardship and societal impact.

In the rest of the paper, the proposed research approach is developed, drawing inspiration from Büyüközkan and Havle [25] and making necessary modifications. The methodology encompasses several key components, beginning with SCOC (Strengths, Challenges, Opportunities, Concerns) analysis, which offers a positive reframing of traditional strategic planning tools, emphasizing solution-oriented thinking. Fuzzy set theory is then introduced to handle uncertainty and imprecision in decision-making processes, facilitating the representation and manipulation of vague or subjective data through fuzzy numbers and operations. Additionally, the Z-Numbers Theory is incorporated to address unreliable numerical computations, providing a framework for handling uncertainty and establishing probability constraints on decision variables. The methodology also includes the Fuzzy Best–Worst Method (BWM), a multi-criteria decision-making approach that integrates fuzzy numbers to evaluate and prioritize alternatives based on subjective judgments and uncertainties associated with decision criteria. Finally, the Z-MARCOS Method is employed to determine the utility functions of options, considering both fuzzy values and reliability for each criterion. Through these methodological approaches, the research aims to comprehensively evaluate and prioritize strategies for sustainable development, taking into account various uncertainties and subjective assessments inherent in decision-making processes.

## 2. Literature Review

### 2.1. Sustainable and Green Hospitality

Before examining studies related to green hospitality, mentioning some valuable and unique examples in the field of sustainability can show the importance of sustainability, environmental management, social responsibilities, and governance. Swedish furniture manufacturer IKEA has one of the most sustainable supply chains, with 50% of its wood sourced from sustainable foresters and 100% of its cotton sourced from sustainable farms that meet standards for wood and cotton exploitation. Additionally, IKEA stores have more than 700,000 solar panels [26].

A notable feature of sustainability at Panasonic Electronics Co. is that it has moved its North American headquarters from suburban New Jersey to Leadership in Energy and Environmental Design (LEED Green Buildings)-certified buildings in downtown Newark. This action by Panasonic reduces the need for employees to drive to work, thereby reducing carbon emissions [27,28].

This interdisciplinary study explores the significance of implementing sustainable practices and effective facility management in the context of establishing a sustainable and responsible hospitality business. The findings highlight the paramount importance placed on sustainable initiatives and resource management, with over 90% of the respondents confirming the significance of managing resources such as food, water, energy, and waste. However, the research also uncovers a contrasting viewpoint, revealing that only 52.9% of the participants would be willing to pay higher prices for sustainable solutions in hotel accommodations. The results and recommendations derived from this study contribute valuable insights by recognizing the competitive advantage of hotels that incorporate sustainable technologies into their business systems and operations through interdisciplinary approaches. Furthermore, this research enhances our understanding of the economic and marketing value of sustainable innovations supported by digital technologies when effectively communicated [29].

In a separate study, an examination was conducted into sustainability management within the hotel industry with a focus on green restaurants. The findings from this research reveal that both hedonic and utilitarian values have a notably positive impact on consumers' inclinations towards green restaurants. Furthermore, utilitarian values and customer preferences exhibit a significant and positive influence on the intentions of individuals to

engage in sustainable behaviors at these green restaurants. Notably, the study's outcomes also indicate that consumer preferences play a partial mediating role in the connections between utilitarian values and the intentions of patrons to engage in sustainable practices at green restaurants [30].

The research by Yusoff and Nejati [31] aims to enhance comprehension of the determinants of green Human Resource Management (HRM), which can potentially result in enhanced environmental and financial performance within the hospitality sector. Their model is constructed by integrating external influences, such as normative pressures, and internal factors, such as the environmental concerns of managers. This integration is achieved by drawing on insights from institutional theory and the theory of the resource-based perspective. The study also introduces a model designed for organizations within the hospitality industry. This model is formulated based on an analysis of sustainability practices within the industry, focusing on the enhancement of both financial and environmental performance, as well as the elucidation of the mechanisms contributing to a company's sustainability.

Lin [32] has delved into the critical role of the water–food–energy (WFE) nexus in the pursuit of sustainable development. Within this investigation, Lin has methodically examined the concept of a WFE link. Lin's findings indicate a lack of comprehensive understanding concerning the reciprocal interactions between the WFE link and its evolution in existing research. Consequently, the following five priority areas for future research are proposed: the establishment of a multi-source database for the WFE, the demonstration of the mutual feedback mechanism inherent in the WFE link, the development of a coupling model for the WFE link, the creation of a decision-making platform dedicated to the WFE link, and the promotion of cooperation among various stakeholders associated with the WFE link. These initiatives are envisioned to facilitate the achievement of sustainable development through the synergetic governance and scientific management of the WFE link. Goh and Muskat's [33] study investigated hospitality students' general attitudes and perceptions about green and sustainable practices; it shows that the need for sustainability and the use of green management methods for hotels is increasing day by day, and this issue can even create potential positive effects in the new generation of graduates by creating relevant scientific content in higher education at universities.

*2.2. Performance Evaluations in Hospitality*

In a recently published study, the significance of Green Management Practices (GMPs) is explored not only as a means of enhancing the overall performance of organizations in environmental, economic, and social aspects but also as a source of competitive advantage. The study's findings, obtained through the application of the Smart PLS structural equation modeling method and an analysis of data from 304 middle managers in small and medium-sized hotels and travel agencies, underscore the need for small and medium-sized enterprises in the hospitality sector to prioritize the establishment of a culture centered on environmental responsibility. Additionally, the findings highlights the importance of encouraging employees to actively participate in green initiatives, thereby promoting sustainability within the industry. This research holds significance as it offers insights into the role of employees' pro-environmental conduct in the realm of green management and sustainable performance within small and medium-sized hospitality enterprises. Its implications are both theoretical and practical, shedding light on the potential for the industry to adopt more sustainable practices while also indicating avenues for future research [34].

The results of the studies of Shah and Ahmed [35] regarding green perspectives and approaches and green intellectual capital show a significant relationship of green people and relational capital with environmental performance. It also shows a strong relationship between environmental responsibility and environmental performance. In Mubeen and Nisar's [36] research concerning the influence of green dynamic capabilities and green practices on the sustainable performance of small and medium-sized enterprises (SMEs), with a specific emphasis on the pivotal roles of green value creation and green innovation,

the findings indicate a significant relationship between green dynamic capabilities and green practices and green innovation, mediated by green value co-creation. This study offers valuable insights to SME practitioners, emphasizing the significance of integrating green innovation and green value creation into their business strategies as a means to improve sustainable performance.

## 3. Materials and Methods

The methodology of the current research is an inspiration from a study conducted by Büyüközkan and Havle [25] with some modifications. The steps of the proposed methodology in the current research are elaborated further.

### 3.1. SCOC Analysis

The evolution of traditional strategic planning tools has given rise to the asset-based SCOC (Strengths, Challenges, Opportunities, Concerns) analysis [37], offering a shift towards a more open and solution-oriented mindset for achieving outstanding results. In contrast to the well-known SWOT analysis [38], which focuses on internal Strengths and Weaknesses, as well as external Opportunities and Threats, SCOC introduces a positive reframing by translating Weaknesses and Threats into Challenges, both internal and external. Reflecting on experiences with SWOT in strategy meetings, it becomes evident that the emphasis on weaknesses often leads to a deficit-based approach, diverting attention to external threats that are often beyond immediate influence [39]. Negative emotions associated with weaknesses and threats tend to overshadow the positive aspects, limiting the ability to recognize opportunities and strengths. Recognizing the psychological impact of negativity, SCOC offers an alternative perspective by categorizing Weaknesses and Threats as Challenges, encouraging a more solution-driven mindset [40].

Comparatively, SCOC aligns with the principles of positive psychology and strategic planning tools such as Strengths, Opportunities, Aspirations, and Results (SOAR) [41], which emphasize strength-based approaches. In contrast to SWOT, SOAR excludes Weaknesses and Threats from its frameworks, aiming to create an alignment of strengths and render weaknesses irrelevant. SCOC, however, acknowledges the importance of Challenges but reframes them to foster a more constructive approach. With SCOC, the analysis takes an asset-based approach, focusing on what works well (Strengths) and how these strengths can be leveraged to address internal Challenges [42]. The framework then explores Opportunities, aligning them with existing strengths, and considers external Challenges with an open and positive mindset. This positive reframing encourages curiosity and creativity, allowing for strategic discussions that are more solution-oriented. While SCOC is not presented as a comprehensive strategic planning framework similar to SWOT, it serves as a valuable tool for analysis and preparation in strategic decision-making [43]. It emphasizes the need for a broader strategic planning process embedded in positive and mindful leadership, where purpose, appreciation, and diversity play central roles. By integrating SCOC into the strategic planning process, organizations can foster a culture that promotes solution-driven thinking and capitalizes on existing strengths for future success.

### 3.2. Fuzzy Set Theory

A fuzzy set is defined as a membership function indicating the degree of membership of elements in a specified range [44], typically represented as the interval [0, 1]. Subsequently, fundamental definitions for fuzzy number sets used in this study are presented. A fuzzy set $A$ defined in the reference $X$ is represented by Equation (1).

$$A = \{(x, \mu_A) | x \in X\} \tag{1}$$

Here, $\mu_A(x) : X \to [0, 1]$ is the membership function of the set $A$. The membership value $\mu_A(x)$ indicates the degree of belonging of $x \in X$ to $A$. A triangular fuzzy number $\widetilde{A}$ is defined as a triple $(l, m, u)$, and its membership function is given by Equation (2).

$$\mu_A(x) = \begin{cases} 0 & x \in (-\infty, l) \\ \frac{x-1}{m-1} & x \in [l, m] \\ \frac{u-x}{u-m} & x \in [m, u] \\ 0 & x \in (u, \infty) \end{cases} \tag{2}$$

Suppose $\widetilde{A} = (l_1, m_1, u_1)$ and $\widetilde{B} = (l_2, m_2, u_2)$ are two triangular fuzzy numbers and $\gamma$ is a constant greater than zero [45]. In this case, arithmetic operations on these fuzzy numbers are performed according to the Equations (3)–(7).

$$\widetilde{A} \oplus \widetilde{B} = (l_1 + l_2, m_1 + m_2, u_1 + u_2) \tag{3}$$

$$\widetilde{A} \otimes \widetilde{B} = (l_1 l_2, m_1 m_2, u_1 u_2) \tag{4}$$

$$\widetilde{A} - \widetilde{B} = (l_1 - u_2, m_1 - m_2, u_1 - l_2) \tag{5}$$

$$\widetilde{A} / \widetilde{B} = (l_1 / u_2, m_1 / m_2, u_1 / l_2) \tag{6}$$

$$\gamma \widetilde{A} = \gamma(l_1, m_1, u_1) = (\gamma l_1, \gamma m_1, \gamma u_1) \tag{7}$$

Suppose $\widetilde{A} = (l_1, m_1, u_1)$ and $\widetilde{B} = (l_2, m_2, u_2)$ are two triangular fuzzy numbers. The distance between $\widetilde{A}$ and $\widetilde{B}$ is defined as follows.

$$d\left(\widetilde{A}, \widetilde{B}\right) = \sqrt{1/3(l_1 - l_2)^2 + (m_1 - m_2)^2 + (u_1 - u_2)^2} \tag{8}$$

### 3.3. Z-Numbers Theory

The concept of Z-numbers was first introduced as a generalization of the theory of uncertainty to handle unreliable numerical computations [46]. Z-numbers, denoted as Z = (A, B), consist of a pair of fuzzy numbers, where the first component A is a fuzzy subset of the domain $X$, and the second component $B$ is a fuzzy subset of the unit interval, indicating the reliability of component $A$. The triplet $(X, A, B)$ is recognized as a Z-valuation, representing an equivalence with a proposition and serving as a general constraint on $X$, defined by Equation (9).

$$Prob~(X~is~A)~is~B \tag{9}$$

This general limit is known as a probability limit that represents a probability distribution function $R(x)$. Specifically, it can be described as Equation (10).

$$R(x) : X~is~\to~poss~(X = u) = \mu_A(u) \tag{10}$$

In the above equation, $_A$ is a membership function of $A$ and $u$ is a general value of $X$. $\mu_A$ can therefore be considered a constraint associated with $R(x)$. This means that $_A(u)$ covers what degree of satisfaction u. Therefore, $X$ is a random variable with probability distribution $R(x)$, which acts as a probability constraint on $X$. The probability limit and the probability density function of $X$ are as described in Equations (11) and (12):

$$R(x) : X~is~p \tag{11}$$

$$R(x) : X~is~p~\to~(u \le X \le u + du) = p(u)~du \tag{12}$$

In Equation (12), *du* indicates the partial derivative of *u*.

### 3.4. Fuzzy Best–Worst Method

In the Fuzzy Best–Worst Method (BWM), a multi-criteria decision-making approach is employed to evaluate and prioritize alternatives based on a set of criteria [47]. This method extends the traditional BWM to handle uncertainty and imprecision in decision-making processes. The core idea behind the Fuzzy BWM is to incorporate fuzzy numbers to represent subjective judgments and uncertainties associated with the decision-making criteria. The process begins with the identification of criteria relevant to the decision context. Decision makers then provide their subjective assessments in the form of fuzzy numbers, capturing the vagueness inherent in their judgments. These fuzzy numbers express the perceived importance or preference for each criterion in relation to others.

The next step involves constructing the Fuzzy Decision Matrix, where each row corresponds to a criterion and the columns represent alternatives [48]. The matrix entries are filled with the fuzzy numbers provided by decision makers. The pairwise comparisons are performed by assessing the relative importance or preference between each criterion in comparison to others. The Fuzzy Best–Worst Index (FBWI) and Fuzzy Worst–Best Index (FWBI) are calculated based on the aggregated preferences obtained from the pairwise comparisons. These indices help determine the best and worst criteria, respectively, indicating their relative significance in the decision-making process. The Fuzzy Relative Proximity to the Best (FRP_B) is then computed to quantify the degree of proximity of each criterion to the best one. The final step involves deriving the Fuzzy Decision Matrix for Alternatives (FDMA) by aggregating the fuzzy numbers associated with each alternative across all criteria. This matrix facilitates the ranking of alternatives based on their overall performance concerning the fuzzy criteria.

### 3.5. Z-MARCOS Method

Multi-criteria Decision Analysis for Complex Systems (MARCOS) has been applied to solve a wide range of different decision-making problems in various studies [49,50]. The MARCOS method determines the utility functions of options by defining the relationship between options and the ideal and anti-ideal points as reference points, and it obtains a compromise ranking of options. Some advantages of the MARCOS method compared to other decision-making methods such as Simple Additive Weighting (SAW), Weighted Aggregated Sum Product Assessment (WASPAS), Technique for Order of Preference by Similarity to Ideal Solution (TOPSIS), etc. [51], include higher efficiency, ease of structuring, optimization of the decision-making process, more accurate determination of desirability in relation to the reference point, stability, stronger results in changing measurement scales conditions, and the absence of rank-inversion problems.

The first step in all multi-criteria decision-making techniques aiming at ranking is the formation of the decision matrix. In the MARCOS technique, evaluation of m options is carried out using n criteria, and each option is scored based on each criterion. Let $A_i$ represent our options and $C_j$ represent the criteria of interest. Therefore, the decision matrix is initially formed with elements of Z values as defined in Equation (13).

$$X = \begin{matrix} & \begin{matrix} C_1 & C_2 & \ldots & C_n \end{matrix} \\ \begin{matrix} A_1 \\ A_2 \\ \ldots \\ A_m \end{matrix} & \begin{bmatrix} x_{11} & x_{12} & \ldots & x_{1n} \\ x_{21} & x_{22} & \ldots & x_{2n} \\ \ldots & \ldots & \ldots & \ldots \\ x_{m1} & x_{m2} & \ldots & x_{mn} \end{bmatrix} \end{matrix} \tag{13}$$

In the 2nd step, the Z values obtained from the decision matrix in the first step are transformed into triangular fuzzy numbers based on Appendix A, and a decision matrix with elements of triangular fuzzy numbers is obtained.

$$
\widetilde{X} = \begin{array}{c} \\ A_1 \\ A_2 \\ \cdots \\ A_m \end{array}
\begin{array}{cccc}
C_1 & C_2 & \cdots & C_n \\
\left[ \begin{array}{cccc}
\left( x_{11}^l, x_{11}^m, x_{11}^u \right) & \left( x_{12}^l, x_{12}^m, x_{12}^u \right) \cdots & \left( x_{1n}^l, x_{1n}^m, x_{1n}^u \right) \\
\left( x_{21}^l, x_{21}^m, x_{21}^u \right) & \left( x_{22}^l, x_{22}^m, x_{22}^u \right) \cdots & \left( x_{2n}^l, x_{2n}^m, x_{2n}^u \right) \\
\cdots & \cdots & \cdots & \cdots \\
\left( x_{m1}^l, x_{m1}^m, x_{m1}^u \right) & \left( x_{m2}^l, x_{m2}^m, x_{m2}^u \right) \cdots & \left( x_{mn}^l, x_{mn}^m, x_{mn}^u \right)
\end{array} \right]
\end{array} \tag{14}
$$

In the 3rd step, based on the Equations (15) and (16), the ideal values ($A_{id}$) and anti-ideal values ($A_{ai}$) are determined.

$$
A_{ai} = \min_{1 \le i \le m} x_{ij}, \ j \in B^{max}, \quad \max_{1 \le i \le m} x_{ij}, \ j \in C^{min} \tag{15}
$$

$$
A_{id} = \max_{1 \le i \le m} x_{ij}, \ j \in B^{max}, \quad \min_{1 \le i \le m} x_{ij}, \ j \in C^{min} \tag{16}
$$

The expression $B$ refers to the criteria with a benefit aspect, and the expression C denotes the criteria with a cost aspect.

In step 4, normalization of the decision matrix takes place. Using Equation (18), normalization is performed for benefit-type criteria, and with Equation (17), normalization is carried out for cost-type criteria.

$$
n_{ij} = \frac{x_{id}}{x_{ij}} \ if \ \ j \in C \tag{17}
$$

$$
n_{ij} = \frac{x_{ij}}{x_{id}} \ if \ \ j \in B \tag{18}
$$

The normalized matrix is multiplied by the criterion weights using Equation (19) to obtain the normalized weighted matrix.

$$
v_{ij} = n_{ij} * w_j \tag{19}
$$

In the next step, the degrees of ideal $K_i^+$ and counter-ideal $K_i^-$ options are calculated using Equations (20) and (21).

$$
K_i^- = \frac{S_i}{S_{ai}} \tag{20}
$$

$$
K_i^+ = \frac{S_i}{S_{id}} \tag{21}
$$

In the above equation, $S_i$ is the sum of the values of each row in the weighted matrix, which is obtained from the following equation.

$$
S_i = \sum_{i=1}^{n} v_{ij} \tag{22}
$$

Finally, the overall desirability of each option is calculated using Equation (23).

$$
f(k_i) = \frac{K_i^+ + K_i^-}{1 + \frac{1 - f\left( K_i^+ \right)}{f\left( K_i^+ \right)} + \frac{1 - f\left( K_i^- \right)}{f\left( K_i^- \right)}} \tag{23}
$$

In the above equation, $f\left( K_i^- \right)$ is the anti-ideal utility function and $f\left( K_i^+ \right)$ is the ideal utility function for the infinite which is calculated from the Equations (24) and (25). Then,

based on the numbers obtained from $f(k_i)$ of each option, the ranking is generated in descending order.

$$f(K_i^-) = \frac{K_i^+}{K_i^+ + K_i^-} \tag{24}$$

$$f(K_i^+) = \frac{K_i^-}{K_i^+ + K_i^-} \tag{25}$$

### 3.6. Proposed Methodology

In this section, the proposed approach of this research is presented, utilizing SCOC, Z-SWARA, and Z-MARCOS methods for evaluating the prioritization of strategies. The proposed approach is presented in three phases. In the first phase, this approach involves the identification of strengths, weaknesses, opportunities, and threats by the SCOC analysis team. Proposed strategies are then scored by the expert team. In this phase, the reliability of each of the strategies extracted from the SCOC analysis is determined by the respective expert team.

In the second phase, to consider different importance for criteria, the Fuzzy BWM method is used. After prioritizing the criteria based on importance using linguistic variables, these variables are transformed into triangular fuzzy numbers. Subsequently, the steps of the Fuzzy BWM method are executed based on these values, and optimal weights for the indices are determined.

In the third phase, based on the outputs of the first and second phases, an attempt is made to prioritize the extracted strategies considering the different importance of indices using the Z-MARCOS method. Unlike the conventional MARCOS method, this method, in addition to considering fuzzy values, has the ability to take into account the reliability for each criterion for each option in this research. After determining the decision matrix, consisting of fuzzy numbers and reliability values (Z-numbers), these values are transformed into triangular fuzzy numbers using Appendix A. Then, the steps of the MARCOS method are executed in a fuzzy environment.

### 4. Results

#### 4.1. Case Study

In Iran, the hotel industry has experienced growth, as evident in the chart depicting the number of standard hotels categorized by star ratings from 2017 to 2022 (Figure 3). However, a notable observation is that only three percent of the total hotels fall within the five- and four-star categories, with approximately thirty percent having obtained standardized certifications. We chose to focus our research specifically on four- and five-star hotels due to their typically larger scale and resources, which make them better suited for implementing and maintaining sustainable practices. These higher-rated hotels often have a greater potential to influence industry standards and set an example for sustainability initiatives within the hospitality sector. By concentrating our efforts on these establishments, we aimed to explore the significant impact that sustainable practices can have on both environmental conservation and guest satisfaction in upscale accommodations. The need for attention in sustainable hotel practices in Iran, with a focus on energy, water, and food consumption and production, becomes evident. Although there are around 1038 hotels in the country, only a limited number have achieved standardized certification, emphasizing the importance of implementing sustainable practices across the industry.

In terms of the food–water–energy nexus, attention should be directed towards the significant water and energy consumption in hotel kitchens for food preparation and service. Moreover, the energy-intensive nature of hotel operations, including HVAC systems and guest room amenities, highlights the need for energy-efficient technologies and practices. Integrating renewable energy sources can be a strategic move to reduce the environmental impact of energy consumption. Concerns regarding waste generation and management in hotels also arise, especially in the context of food consumption and packaging mate-

rials. Implementing waste-reduction initiatives and sustainable sourcing practices for food can contribute to a more environmentally friendly hotel industry. The data on the distribution of internal travels in Iran across provinces indicate that Khorasan Razavi, Mazandaran (Figure 3a), and Tehran have the highest numbers of trips and, accordingly, hotels. Sustainable practices should be prioritized in these regions due to the concentration of hotel establishments.

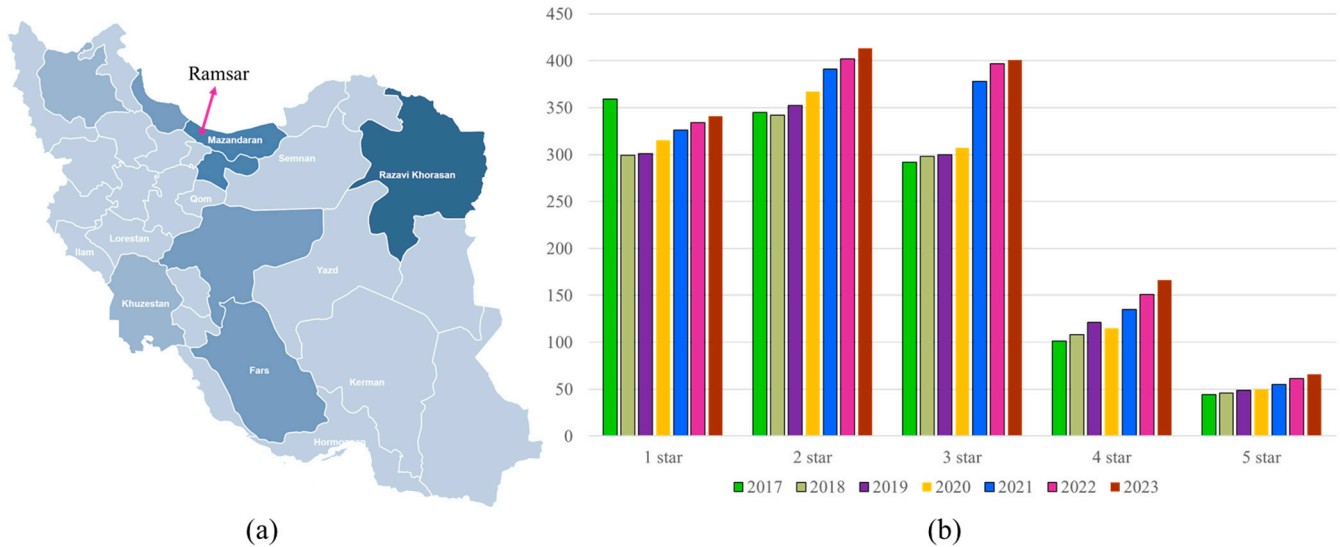

(a)                                                           (b)

**Figure 3.** (**a**) Heatmap of number of internal trips from 2013 to 2023 by province and the location of Ramsar. (**b**) Numbers of 1-star to 5-star hotels in Iran from 2017 to 2022 (source: Statistical Center of Iran).

In this regard, Ramsar, located in Mazandaran (Figure 3a), experiences rainfall throughout the year, with extreme seasonal variation (Figure 4a). The month with the highest average rainfall is October, with 118.4 mm, while the least rainy month is June, with an average of 14 mm. This seasonal rainfall pattern provides an opportunity for effective water management in the region. In terms of solar energy, Ramsar receives varying levels throughout the year. The brightest period spans from 9 May to 19 August, with an average daily incident shortwave energy above 6.8 kWh per square meter. June stands out as the brightest month, with an average of 7.8 kWh. Conversely, the darker period extends from 28 October to 12 February, with an average daily incident shortwave energy below 3.5 kWh per square meter. December is the darkest month, with an average of 2.4 kWh.

Given the considerable sunlight availability, there is significant potential for the use of solar energy in Ramsar (Figure 4b), especially for large buildings. Rooftop solar panels can harness this energy to generate electricity, contributing to sustainable practices and reducing reliance on conventional energy sources. The varying daylight hours and solar energy levels throughout the year provide an opportunity for comprehensive energy planning, allowing buildings to optimize their solar energy utilization.

In addition to solar energy utilization, the region's rainy climate offers possibilities for implementing green roofs. These roofs, covered with vegetation, not only enhance energy efficiency but also mitigate stormwater runoff, acting as natural insulators and promoting biodiversity. The inflow and overflow of water from roofs can be effectively managed through the installation of rainwater harvesting systems. Reservoirs can store rainwater for later use, addressing water scarcity concerns during drier periods. This integrated approach, combining solar energy utilization with sustainable water management practices, aligns with the broader goals of environmental conservation and resilient urban development in Ramsar.

Ramsar also presents a favorable case for integrating innovative technologies to enhance sustainability in hotel construction. Notably, the incorporation of rainwater harvest-

ing systems on buildings with green roofs stands out as a promising solution. Insights from a study in Pyongyang highlight the effectiveness of this technology in optimizing water management, emphasizing the significance of an increased catchment surface for enhanced reliability. Lessons from Malaysia's green hospitality industry underscore the need for improvement in rainwater harvesting systems within Ramsar's hotel industry.

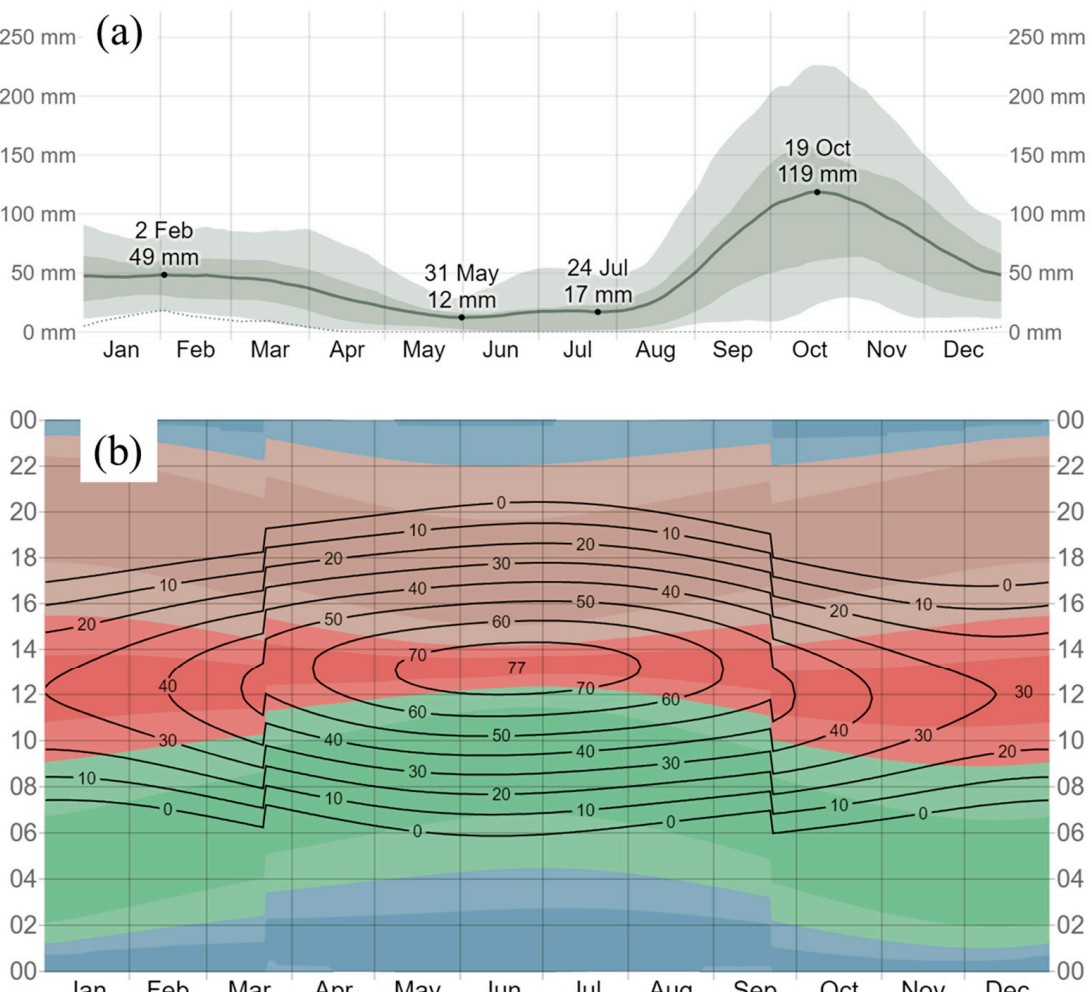

**Figure 4.** (**a**) Depicts the average rainfall (solid line) accumulated over a sliding 31-day period centered on the day in question, with 25th to 75th and 10th to 90th percentile bands. The corresponding average snowfall is represented by the thin dotted line. (**b**) Illustrates solar elevation and azimuth throughout the year 2023. Black lines indicate constant solar elevation (sun angle above the horizon, in degrees), while background color fills represent the azimuth (compass bearing) of the sun. Lightly tinted areas at cardinal compass points indicate implied intermediate directions (northeast, southeast, southwest, and northwest).

Additionally, the introduction of a hydroponic green roof system (HGRS) offers an innovative approach to urban stormwater management, reducing runoff and treating gray water and rainwater on site. The potential for Ramsar to integrate this system aligns with water management and sustainable building practices. Exploring green roofs for urban farming, inspired by Hong Kong's experience, provides a viable Low-Impact Development (LID) technique for Ramsar's high-density urban areas, promoting environmental, social, and economic sustainability. Leveraging the benefits of green roofing systems, including flood risk reduction and energy savings, aligns with Ramsar's water conservation goals. Furthermore, assessing green roof technology within established green building rating systems is crucial for guiding sustainable hotel construction in Ramsar. In summary,

Ramsar's potential to implement these technologies creates an internal circular system addressing the water–energy–food nexus (Figure 5), contributing to eco-friendly hotel construction and broader sustainable urban development goals.

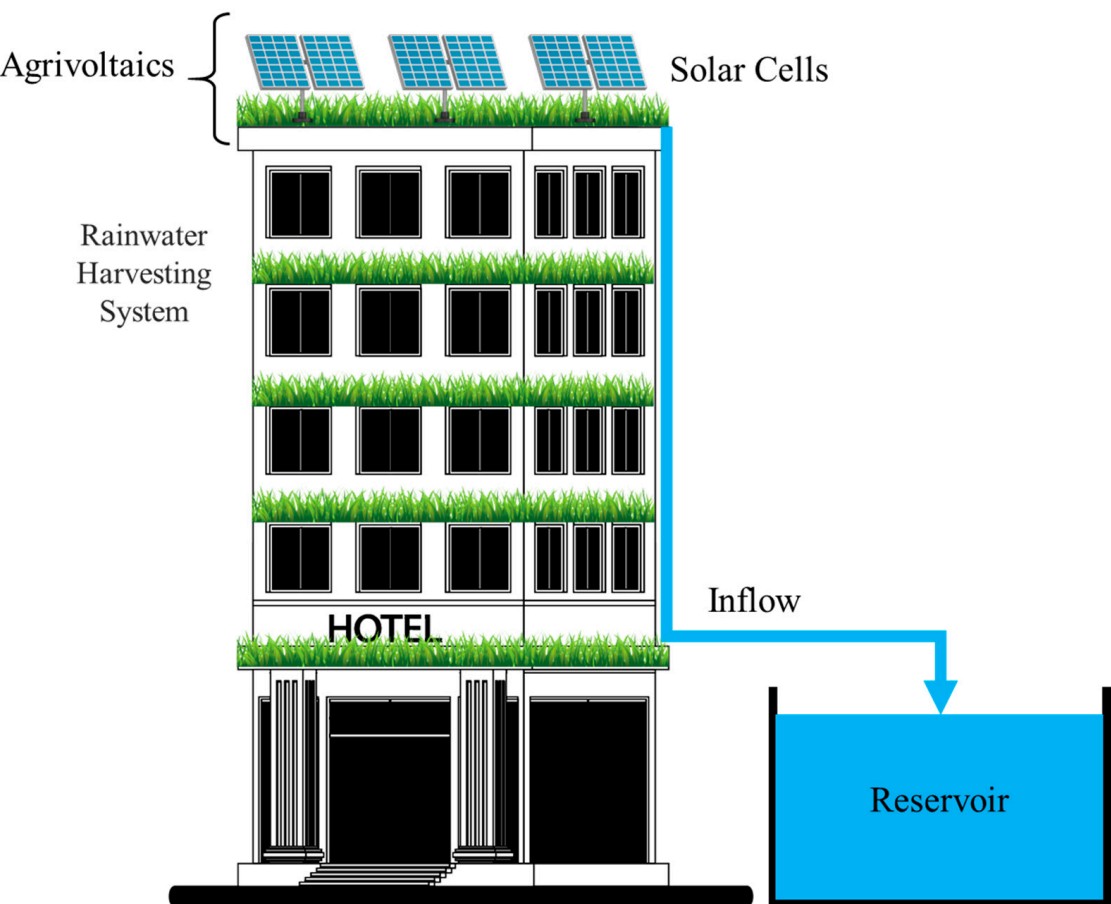

**Figure 5.** Concept of nexus-based hotel design considering sustainable technologies.

### 4.2. SCOC Analysis

We will utilize SCOC analysis in various stages of strategic planning. SCOC analysis categorizes strategies into four groups (Table 1), showcasing sound strategic directions. The objective of this research is to identify sustainable hotel implementation strategies based on the previously discussed nexus criteria. This section outlines how a hotel can meet nexus criteria and what actions stakeholders should take in this regard. To achieve this, seven experts in the fields of hospitality, environmental studies, technology, and industrial engineering, possessing knowledge of the hotel industry, convened. In a focused group session, they discussed and examined the topic. Each expert had more than 5 years of experience in the hotel industry, with a minimum of a master's degree. The summary of expert opinions (Table 1) is as follows.

The criteria for prioritizing top strategies encompass the following considerations. Implementation (IM) assesses the ease of implementing each strategy, examining its practical feasibility and straightforwardness in execution. Water–Food Efficiency (WF) concentrates on the effectiveness of strategies in optimizing the water and food aspects of the system, evaluating their contribution to efficient water and food resource management. Water–Energy Efficiency (WE) examines the effectiveness of strategies in the integrated use of water and energy resources, analyzing their contribution to optimizing water and energy utilization. Energy–Food Efficiency (EF) evaluates strategies' effectiveness in achieving efficiency in the energy and food dimensions, exploring how well they enhance synergy between energy and food components. Estimated Implementation Cost (CO) considers

the projected financial expenses associated with executing each strategy, evaluating the expected costs involved in their implementation.

**Table 1.** Strategies extracted from SCOC analysis.

| SCOC Quadrant | Strategy | Code | Description/Activity |
|---|---|---|---|
| Internal Strengths: | Operationalizing hydroponic green roof system implementation | IS1 | Develop a phased implementation plan with clear milestones. Integrate hydroponic systems into existing roof infrastructure efficiently. Regularly monitor and optimize hydroponic performance for sustainability. |
| | Embedding green approaches in hotel operations | IS2 | Conduct a comprehensive audit of current operations for eco-friendly opportunities. Introduce sustainable practices into daily hotel routines and procedures. Train staff to embrace and champion green initiatives. |
| | Leveraging local expertise for sustainable hotel practices | IS3 | Collaborate with local experts to identify region-specific sustainable practices. Establish partnerships with local suppliers for sustainable sourcing. |
| Internal Challenges: | Mitigating financial constraints through strategic planning | IC1 | Conduct a cost-benefit analysis for sustainable initiatives. Explore governmental financing options and partnerships for sustainable investments. |
| | Bridging skill and knowledge gaps for effective implementation | IC2 | Provide ongoing training programs for staff on sustainable practices. Collaborate with educational institutions for specialized training. |
| | Overcoming infrastructure limitations for sustainable infrastructure | IC3 | Identify and address infrastructure gaps hindering sustainability. Explore modular and adaptive infrastructure solutions. |
| External Opportunities: | Strategically promoting sustainable tourism initiatives | EO1 | Collaborate with local tourism boards to promote sustainable attractions. Engage with guests to raise awareness and encourage sustainable behaviors. |
| | Harnessing government support for green hotel development | EO2 | Advocate for green incentives and policies at the local and national levels. Stay informed about and comply with evolving environmental regulations. |
| | Facilitating international collaboration for best practices | EO3 | Exchange best practices with global hospitality networks. Implement successful strategies from other regions where applicable. |
| | Establishing strategic technology partnerships for innovation | EO4 | Collaborate with technology providers for sustainable solutions. Invest in cutting-edge technologies to enhance energy efficiency. Regularly evaluate and update technology partnerships for continuous improvement. |
| | Pursuing green certification recognition for hotels | EO5 | Undertake the necessary steps to achieve recognized green certifications. Showcase certifications in marketing materials to attract environmentally conscious guests. Maintain compliance with certification requirements through regular assessments. |
| External Challenges: | Navigating political stability challenges for sustainable goals | EC1 | Develop contingency plans for potential political disruptions. |
| | Addressing climate change risks in hotel design and operations | EC2 | Conduct climate risk assessments for both current and future scenarios. Integrate climate-resilient design elements into the hotel's infrastructure. Develop emergency response plans to mitigate the impact of extreme weather events. |
| | Adapting to evolving legal and regulatory environment | EC3 | Stay informed about changes in environmental laws and regulations. Adapt hotel operations promptly to comply with evolving legal standards. |
| | Staying technologically relevant, minimizing obsolescence risk | EC4 | Regularly assess and upgrade technological systems to stay current. Invest in scalable technologies that can adapt to future advancements. |

### 4.3. Results Analysis

In this section, the outcomes of implementing the proposed research approach in evaluating hotel development strategies, considering the establishment of a water–energy–food nexus system, are presented and scrutinized. Following the methodology's initial phase, the SCOC analysis team determines index values for each strategy. Given the uncertainties associated with these factors, the Z-number theory is employed. In this

theory, besides considering the fuzzy nature of the indices, their reliability is also taken into account. The Z-values for the indices concerning strategies are presented in Table 2.

**Table 2.** The index values for strategies in terms of Z-numbers.

| Strategy | IM | WF | WE | EF | CO |
|----------|-----|------|------|------|------|
| IS1 | FB, F | F, E | F, B | FG, G | FB, F |
| IS2 | FB, E | FG, B | F, G | G, W | FB, E |
| IS3 | F, B | FG, G | FG, W | FG, B | F, B |
| IC1 | F, G | G, W | FG, B | FG, G | F, G |
| IC2 | FG, W | FB, W | FG, B | G, W | FG, B |
| IC3 | FG, F | FB, F | FG, G | FB, W | FG, G |
| EO1 | FG, E | FB, E | G, W | FB, F | G, W |
| EO2 | G, B | F, B | FB, W | FB, E | FB, W |
| EO3 | F, F | G, W | FB, E | F, B | FG, B |
| EO4 | F, E | FB, W | F, B | F, G | FG, G |
| EO5 | FG, B | FB, F | F, G | FG, W | G, W |
| EC1 | FG, G | FB, E | FB, W | F, F | FB, E |
| EC2 | G, W | F, B | FB, F | F, E | G, W |
| EC3 | FB, W | F, G | FB, E | FG, B | FB, W |
| EC4 | G, G | F, G | FB, F | F, B | FB, F |

Table 2 displays the index values for strategies in terms of Z-numbers. This analysis contributes to a comprehensive understanding of the prioritization of strategies, taking into account the given criteria and utilizing the Z-number theory for handling uncertainties and reliability considerations.

The priority ranking of criteria based on expert opinion importance levels is presented. The criteria include Implementation (IM), Water–Food Efficiency (WF), Water–Energy Efficiency (WE), Energy–Food Efficiency (EF), and Cost (CO). The linguistic variables associated with each criterion and their corresponding importance levels, as indicated by the experts, are outlined. For Implementation (IM), the linguistic variables are FB (Fairly Bad) and G (Good), denoting a moderate importance level and a high importance level, respectively. Water–Food Efficiency (WF) is characterized by the linguistic variables FB (Fairly Bad) and F (Fair), suggesting a moderately low importance level and an equal importance level. Water–Energy Efficiency (WE) involves the linguistic variables FG (Fairly good) and E (Excellent), indicating a moderately high importance level and an excellent importance level. Energy–Food Efficiency (EF) is associated with the linguistic variables F (Fair) and B (bad). Lastly, the Cost (CO) criterion is defined by the linguistic variables FB (Fairly Bad) and F (Fair). Then the next steps of the F-BWM method are implemented according to these triangular fuzzy numbers.

According to the Table 3, it can be seen that the weight of the indicators is specified in the form of triangular fuzzy numbers in the last column. In the third phase of the proposed approach and based on the results of the first and second phases, the prioritization of failure states is performed using the developed Z-MARCOS method. At first, the decision matrix of the Z-MARCOS method is generated in the form of Z-numbers (considering uncertainty and reliability), in such a way that the rows of this matrix indicate the evaluated options or the same strategy and the columns of this matrix indicate the evaluation criteria. In the following, the said decision matrix is transformed into a decision matrix in the form of triangular fuzzy numbers, which is presented in Table 4, using the transformations presented in Appendix A. Now, after assigning the number to the linguistic variables and its calculations, the weighted normalized matrix is obtained by considering the weights of the indicators (Appendix B). In this section, the Z-MARCOS method is implemented and its results are presented considering the uncertainty in the indicators and the reliability in the strategies. The results of the Z-MARCOS approach are presented in Table 4.

**Table 3.** The weight of each criterion in fuzzy conditions.

| Criteria | $S_j$ | | | $K_j$ | | | $q_j$ | | | $W_j$ | | |
|---|---|---|---|---|---|---|---|---|---|---|---|---|
| | *l* | *m* | *u* | *l* | *m* | *u* | *l* | *m* | *u* | *l* | *m* | *u* |
| IM | 0.56 | 0.84 | 1.26 | 1.561 | 1.837 | 2.255 | 0.443 | 0.544 | 0.641 | 0.158 | 0.222 | 0.306 |
| WF | 0.38 | 0.47 | 0.64 | 1.379 | 1.474 | 1.636 | 0.271 | 0.369 | 0.465 | 0.097 | 0.15 | 0.222 |
| WE | 0.21 | 0.24 | 0.28 | 1.209 | 1.237 | 1.275 | 0.213 | 0.299 | 0.384 | 0.086 | 0.114 | 0.182 |
| EF | 0 | 0.55 | 1.64 | 1 | 1 | 1 | 1 | 1 | 1 | 0.359 | 0.409 | 0.474 |
| CO | 0.21 | 0.23 | 0.28 | 1.205 | 1.233 | 1.283 | 0.166 | 0.242 | 0.319 | 0.056 | 0.098 | 0.156 |

**Table 4.** Prioritization of strategies based on the Z-MARCOS approach.

| Strategy | f(K+) | | | f(K−) | | | $f_{K+}$ | $f_{K-}$ | K+ | K− | $f(K_i)$ | Rank |
|---|---|---|---|---|---|---|---|---|---|---|---|---|
| | *l* | *m* | *u* | *l* | *m* | *u* | | | | | | |
| IS1 | 0.12001 | 0.49814 | 3.2648 | 0.02004 | 0.07966 | 0.28282 | 0.89697 | 0.10302 | 0.9942 | 8.647 | 0.98161 | 1 |
| IS2 | 0.08184 | 0.34814 | 2.29863 | 0.01422 | 0.05416 | 0.19902 | 0.62996 | 0.07154 | 0.69717 | 6.07384 | 0.46948 | 7 |
| IS3 | 0.0991 | 0.40604 | 2.47791 | 0.01725 | 0.06382 | 0.21501 | 0.70027 | 0.08131 | 0.78778 | 6.7637 | 0.59581 | 4 |
| IC1 | 0.1256 | 0.48916 | 2.83753 | 0.02183 | 0.07679 | 0.24669 | 0.81925 | 0.0955 | 0.92859 | 7.9162 | 0.83208 | 2 |
| IC2 | 0.07707 | 0.30665 | 1.76758 | 0.0146 | 0.04772 | 0.15398 | 0.51147 | 0.0597 | 0.57904 | 4.93745 | 0.31385 | 11 |
| IC3 | 0.0474 | 0.26954 | 1.89837 | 0.00722 | 0.04236 | 0.16479 | 0.50326 | 0.05669 | 0.55141 | 4.86397 | 0.29362 | 12 |
| EO1 | 0.03999 | 0.25663 | 1.75893 | 0.0078 | 0.03978 | 0.15102 | 0.47059 | 0.05401 | 0.51778 | 4.54097 | 0.25567 | 14 |
| EO2 | 0.03571 | 0.24241 | 1.70388 | 0.00511 | 0.03811 | 0.14726 | 0.45137 | 0.05172 | 0.4927 | 4.35315 | 0.23241 | 15 |
| EO3 | 0.06435 | 0.31051 | 1.9972 | 0.01015 | 0.04891 | 0.17286 | 0.5504 | 0.06215 | 0.61209 | 5.31041 | 0.35732 | 8 |
| EO4 | 0.07945 | 0.37109 | 2.33595 | 0.01362 | 0.0597 | 0.20243 | 0.65025 | 0.07496 | 0.725 | 6.27813 | 0.50769 | 6 |
| EO5 | 0.05903 | 0.27656 | 1.76473 | 0.01053 | 0.04344 | 0.15356 | 0.48802 | 0.05676 | 0.54462 | 4.71353 | 0.27956 | 13 |
| EC1 | 0.05799 | 0.30278 | 2.0434 | 0.00923 | 0.04701 | 0.17621 | 0.55011 | 0.06269 | 0.6089 | 5.32392 | 0.35424 | 9 |
| EC2 | 0.08697 | 0.39216 | 2.43501 | 0.0147 | 0.0611 | 0.21108 | 0.68107 | 0.07855 | 0.76215 | 6.58411 | 0.56099 | 5 |
| EC3 | 0.091 | 0.41092 | 3.13988 | 0.01554 | 0.06413 | 0.27165 | 0.81254 | 0.09064 | 0.88127 | 7.83968 | 0.77882 | 3 |
| EC4 | 0.05743 | 0.29327 | 1.97229 | 0.0092 | 0.04684 | 0.17034 | 0.53355 | 0.06149 | 0.59049 | 5.15627 | 0.334 | 10 |

According to the Table 4, it can be seen that based on the Z-MARCOS approach, strategies IS1, IC1, EC3 are placed in the first to third priorities. In other words, these strategies are considered superior strategies and are more capable of being implemented. According to this approach, it can be seen that the EO2 strategy is ranked as the last priority, and due to the limitations, it is not currently a priority for implementation.

## 5. Discussion and Managerial Insights

Although a ranking of strategies has been provided, the analysis of strategies does not conclude here. Upon careful consideration of the criteria used, it becomes evident that IM (Implementation Ease), WF (Water–Food Efficiency), WE (Water–Energy Efficiency), EF (Energy–Food Efficiency), and CO (Estimated Implementation Cost) fall into two categories: IM and CO are of an executive nature, while WF, WE, and EF signify the importance of implementing these strategies. Therefore, it can be inferred that by aggregating similar criteria, an Importance–Performance Analysis (IPA) can be conducted [52]. This analysis holds significant utility for the implementation of options, policies, and strategies. Since the current data allow us to conduct such an analysis, a summary of this analysis will be provided in the discussion section, leading to managerial evaluations and operational insights. The summary table of the IPA is included in Appendix C, and its graph is illustrated in Figure 6.

There are various methods for analyzing the IPA figure, with the most common being to divide the chart into four quadrants using thresholds. However, a vertical division may not be as effective for analysis. Another approach involves using an iso-rating line to create a more effective division of the chart. This results in one large triangle in the top left and one large triangle divided into two smaller triangles and one quadrant. In our analysis, three strategies, namely EC4, EO1, and IC3, fall within the "Concentrate Here"

area, positioned on the borderline. Notably, no strategies are classified under "Keep Up the Good Work". Within the small triangle designated "Low Priority", five strategies—IS1, EC1, EO2, IS3, and EO4—are identified. Additionally, all other strategies are placed in the "Possible Overkill" area, indicating that they are the lowest priority compared to the others.

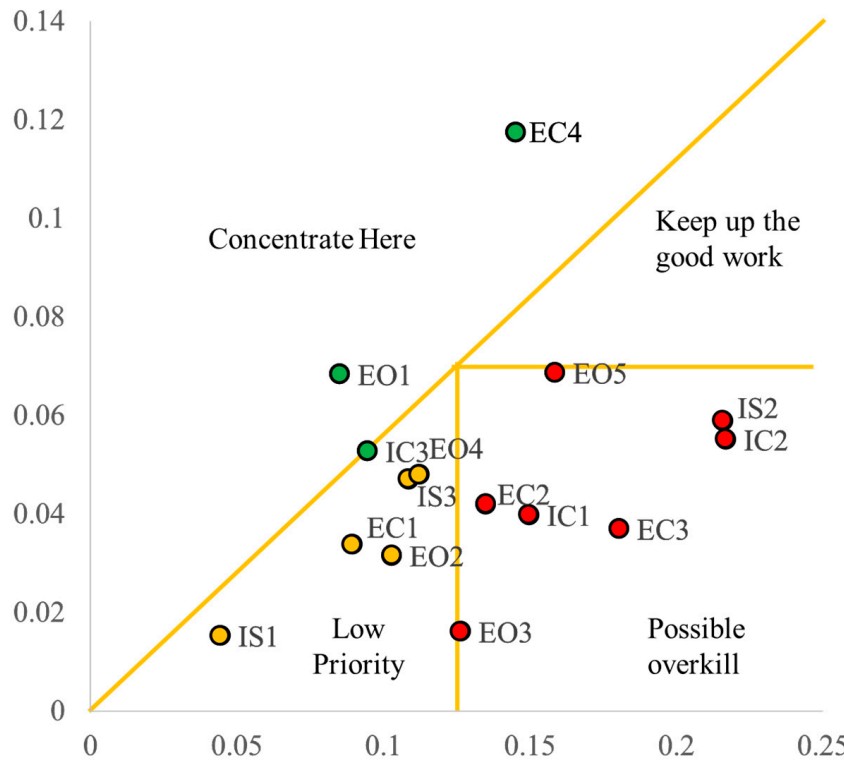

**Figure 6.** The iso-rating-line IPA analysis and coordinates of the strategies.

*Managerial Insights*

Based on the comprehensive analysis conducted, several knowledgeable and managerial insights emerge for the implementation, establishment, and improvement of hotels in Ramsar, Mazandaran, with a focus on agrivoltaics [53], hydropower, solar cells [54], water reservoirs, and green roofs [55] within the context of the energy–water–food nexus. This analysis can guide managerial evaluations and operational decisions. The IPA summary table is provided in Appendix C for reference.

Examining the specific strategies proposed for hotel development in Ramsar, notable considerations emerge. The region's hotel industry has experienced growth, but attention must be given to sustainable practices, particularly in energy, water, and food consumption and production. Only a limited number of hotels have achieved standardized certification, emphasizing the importance of implementing sustainable practices across the industry. The food–water–energy nexus in hotel operations demands a focus on water and energy consumption in kitchens, energy-intensive operations, and waste generation. Implementing waste reduction initiatives and sustainable sourcing practices is crucial for an environmentally friendly hotel industry.

Considering the environmental context of Ramsar, characterized by rainfall and varying solar energy availability, strategic integration of solar energy, green roofs, and water management practices is recommended. Rooftop solar panels can harness ample sunlight, contributing to sustainable practices and reducing reliance on conventional energy sources. The rainy climate offers opportunities for green roofs, which not only enhance energy efficiency but also mitigate stormwater runoff, promote biodiversity, and serve as natural insulators. Rainwater harvesting systems and reservoirs can effectively manage water resources, addressing concerns during drier periods.

Innovative technologies, such as hydroponic green roof systems (HGRS), provide an opportunity for urban stormwater management and on-site treatment of gray water and rainwater. Ramsar's potential to implement these technologies aligns with water management and sustainable building practices, contributing to the eco-friendly construction of hotels and broader sustainable urban development goals. Lessons from successful implementations in other regions, such as Pyongyang and Hong Kong, highlight the effectiveness of these technologies in optimizing water management; reducing runoff; and promoting environmental, social, and economic sustainability. By and large, the integration of agrivoltaics, hydropower, solar cells, water reservoirs, and green roofs in hotel construction in Ramsar presents a holistic approach to address the energy–water–food nexus. These strategies contribute to eco-friendly hotel development, enhance sustainability, and align with Ramsar's broader goals of water conservation and resilient urban development.

## 6. Conclusions

In conclusion, this research addresses critical issues in the hotel industry in Iran, emphasizing the need for sustainable practices, particularly in energy, water, and food consumption and production. The case of Ramsar in Mazandaran underscores the significance of implementing eco-friendly strategies to address the challenges associated with the food–water–energy nexus. Despite the growth in the hotel industry, a limited number of establishments have achieved standardized certification, highlighting the necessity for widespread adoption of sustainable practices across the sector.

Our approach, employing SCOC, Fuzzy BWM, and Z-MARCOS methods, provides a robust framework for evaluating and prioritizing strategies in hotel development. Through the SCOC analysis, the positive reframing of weaknesses into challenges fosters a solution-oriented mindset. The Fuzzy BWM method allows for the incorporation of uncertainty, providing a nuanced evaluation of criteria importance. The Z-MARCOS method optimizes decision making, considering the reliability of criteria in a fuzzy environment.

In the case of Iran, the strategies derived from our analysis focus on addressing water and energy consumption in hotel kitchens, energy-intensive operations, and waste generation. The integration of renewable energy sources, waste reduction initiatives, and sustainable sourcing practices emerges as key solutions. For Ramsar, the potential for solar energy utilization, green roofs, and innovative technologies such as hydroponic green roof systems aligns with the region's environmental context. The integrated approach of combining solar energy, water management, and sustainable practices contributes to eco-friendly hotel construction and broader sustainable urban development goals.

Our study aligns with Sorin and Sivarajah's [56] investigation into the understanding and applicability of the circular economy in the hotel industry, albeit focusing on Iranian hotel challenges and opportunities, particularly regarding sustainable practices related to energy, water, and food. Similarly, Lagioia and Amicarelli [57] explore sustainable and circular practices, specifically in food waste management within the Southern Italian hotel industry. While their research concentrates on attitudes and perceptions of hotel managers in the Apulia region, ours expands to encompass diverse sustainability challenges across Iran. Despite geographical disparities, both studies underscore the importance of adopting eco-friendly strategies to address environmental concerns and enhance operational efficiency within the hospitality sector. These references underscore the global relevance and significance of sustainable practices in the hotel industry, thereby contextualizing our research within the broader landscape of hospitality sustainability initiatives.

In comparison to other hotels or architectural projects in Iran or internationally, the proposed approach aligns well with large-scale and prominent structures such as governmental buildings, military installations, high-rise apartments, and shopping malls. However, it is essential to conduct further investigation into the specific context of these projects, considering factors such as stakeholders, regulatory frameworks, and other relevant variables. While the applicability of such initiatives may vary depending on the project's unique characteristics and external factors, hotels present a particularly conducive environment for

implementing these sustainability measures. This is largely due to the consistent management and ownership structures often found within the hospitality industry, which facilitate the implementation of innovative ideas and initiatives. Consequently, the feasibility and effectiveness of integrating sustainability practices into hotels are more readily apparent and achievable compared to other architectural projects with diverse stakeholder dynamics and regulatory challenges.

Looking ahead, future research should explore alternative Multi-Attribute Decision Making (MADM) methods; versions of rough set theory; and the application of gray numbers for SCOC, SOPA, SOAR, and other tools. Researchers are encouraged to focus on additional green, sustainable, and technical strategies to enhance hotels based on the water–food–energy nexus, promoting circular approaches. Moreover, evaluating the efficiency of the proposed strategies in real-world implementations should be a priority to validate their impact and applicability.

**Author Contributions:** A.T. contributed to the idea creation and provided supervision throughout the project. A.P. and V.E. were involved in idea creation, performed modeling, and conducted the case study. M.K. and S.E. contributed to the writing, manuscript revision, and additional modifications. All authors have read and agreed to the published version of the manuscript.

**Funding:** This research received no external funding.

**Institutional Review Board Statement:** Not applicable.

**Informed Consent Statement:** Not applicable.

**Data Availability Statement:** We have included the relevant data in the manuscript, including tables and appendices. However, in certain instances where the data matrices are exceptionally large and impractical to include in the manuscript, they can be made available upon request.

**Conflicts of Interest:** The authors declare no conflicts of interest.

## Appendix A

**Table A1.** Conversion of Linguistic Variables Related to Z-Numbers to Triangular Fuzzy Numbers.

| Linguistic Variable | Membership Function | | | Linguistic Variable | Membership Function | | |
|---|---|---|---|---|---|---|---|
| | *l* | *m* | *u* | | *l* | *m* | *u* |
| E, E | 8.24 | 8.79 | 9.28 | E, G | 7.53 | 8.49 | 8.49 |
| E, F | 6.74 | 7.07 | 7.07 | E, B | 4.93 | 5.3 | 5.77 |
| E, W | 2.85 | 3.16 | 3.16 | G, E | 6.64 | 8.54 | 9.49 |
| G, G | 5.48 | 7.53 | 8.37 | G, F | 4.95 | 6.36 | 7.31 |
| G, B | 3.29 | 4.93 | 5.48 | G, W | 2.12 | 2.85 | 3.45 |
| FG, E | 4.49 | 6.64 | 8.54 | FG, G | 4.18 | 5.86 | 7.59 |
| FG, F | 3.34 | 4.95 | 6.36 | FG, B | 2.74 | 3.49 | 4.59 |
| FG, W | 1.5 | 2.12 | 2.85 | F, E | 2.85 | 4.74 | 6.23 |
| F, G | 2.94 | 4.28 | 5.86 | F, F | 2.12 | 3.54 | 4.86 |
| F, B | 1.45 | 2.74 | 3.83 | F, W | 0.91 | 1.58 | 2.19 |
| FB, E | 0.98 | 2.85 | 4.74 | FB, G | 0.81 | 2.51 | 4.18 |
| FB, F | 0.69 | 2.19 | 3.54 | FB, B | 0.56 | 1.59 | 2.74 |
| FB, W | 0.34 | 0.91 | 1.58 | B, E | 0 | 0.92 | 2.94 |
| B, G | 0 | 0.84 | 2.51 | B, F | 0 | 0.68 | 2.02 |
| B, B | 0 | 0.55 | 1.64 | B, W | 0 | 0.33 | 0.9 |
| W, E | 0 | 0 | 0.91 | W, G | 0 | 0 | 0.81 |
| W, F | 0 | 0 | 0.69 | W, B | 0 | 0 | 0.51 |
| W, W | 0 | 0 | 0.31 | | | | |

## Appendix B

**Table A2.** Weighted Normalized Decision Matrix.

| Strategy | IM | | | WF | | | WE | | | EF | | | CO | | |
|---|---|---|---|---|---|---|---|---|---|---|---|---|---|---|---|
| | *l* | *m* | *u* | *l* | *m* | *u* | *l* | *m* | *u* | *l* | *m* | *u* | *l* | *m* | *u* |
| IS1 | 0.01462 | 0.03373 | 0.13888 | 0.03722 | 0.09542 | 0.19561 | 0.01664 | 0.04356 | 0.09335 | 0.19724 | 0.31767 | 0.47886 | 0.00699 | 0.02814 | 0.07219 |
| IS2 | 0.01053 | 0.02416 | 0.10292 | 0.0347 | 0.07781 | 0.14434 | 0.02523 | 0.06804 | 0.14256 | 0.10505 | 0.15313 | 0.20114 | 0.00786 | 0.0367 | 0.09541 |
| IS3 | 0.01256 | 0.02583 | 0.05909 | 0.05495 | 0.1178 | 0.22193 | 0.01674 | 0.03619 | 0.06996 | 0.13069 | 0.20735 | 0.31218 | 0.01374 | 0.03517 | 0.07627 |
| IC1 | 0.00926 | 0.01702 | 0.03858 | 0.02755 | 0.05603 | 0.09386 | 0.02855 | 0.06141 | 0.12015 | 0.19827 | 0.31639 | 0.47864 | 0.01957 | 0.05673 | 0.11918 |
| IC2 | 0.01809 | 0.03106 | 0.06204 | 0.00426 | 0.01961 | 0.04722 | 0.02814 | 0.06112 | 0.12026 | 0.10556 | 0.15347 | 0.20155 | 0.02074 | 0.04932 | 0.09995 |
| IC3 | 0.0071 | 0.01411 | 0.02841 | 0.0086 | 0.04241 | 0.10432 | 0.0414 | 0.09569 | 0.18379 | 0.01539 | 0.05139 | 0.10004 | 0.03285 | 0.07767 | 0.15208 |
| EO1 | 0.00609 | 0.01128 | 0.02162 | 0.01116 | 0.05732 | 0.14029 | 0.02231 | 0.04593 | 0.07656 | 0.03359 | 0.11588 | 0.22443 | 0.01679 | 0.03669 | 0.06463 |
| EO2 | 0.00801 | 0.01465 | 0.02553 | 0.02112 | 0.05402 | 0.11219 | 0.00383 | 0.01551 | 0.03964 | 0.04594 | 0.15463 | 0.30071 | 0.00206 | 0.01234 | 0.03281 |
| EO3 | 0.00983 | 0.01937 | 0.04602 | 0.02764 | 0.05668 | 0.09311 | 0.01072 | 0.04508 | 0.11693 | 0.07732 | 0.14872 | 0.24285 | 0.022 | 0.04973 | 0.09965 |
| EO4 | 0.0075 | 0.01517 | 0.03433 | 0.00476 | 0.01932 | 0.04745 | 0.01539 | 0.04325 | 0.09309 | 0.11939 | 0.23287 | 0.37259 | 0.03251 | 0.07782 | 0.15205 |
| EO5 | 0.00954 | 0.01841 | 0.03585 | 0.00958 | 0.04193 | 0.10381 | 0.02518 | 0.06932 | 0.14279 | 0.07454 | 0.11993 | 0.18161 | 0.01698 | 0.03788 | 0.06369 |
| EC1 | 0.00604 | 0.01281 | 0.02243 | 0.01236 | 0.05716 | 0.14056 | 0.00288 | 0.01554 | 0.03834 | 0.10074 | 0.19104 | 0.31416 | 0.00754 | 0.03641 | 0.0953 |
| EC2 | 0.01669 | 0.02452 | 0.04323 | 0.02086 | 0.05474 | 0.11364 | 0.00623 | 0.03424 | 0.0862 | 0.13498 | 0.25627 | 0.42014 | 0.01706 | 0.03758 | 0.0632 |
| EC3 | 0.03238 | 0.07436 | 0.30544 | 0.03126 | 0.0851 | 0.17272 | 0.00963 | 0.04626 | 0.11557 | 0.13065 | 0.20858 | 0.31297 | 0.00349 | 0.0128 | 0.03209 |
| EC4 | 0.00696 | 0.0085 | 0.01736 | 0.03202 | 0.0859 | 0.17258 | 0.00635 | 0.0347 | 0.08646 | 0.07733 | 0.1483 | 0.24273 | 0.00538 | 0.02831 | 0.07141 |

## Appendix C

**Table A3.** The Importance–Performance Values Derived from the Z-MARCOS Results.

| Code | Importance (Fuzzy) | | | Performance (Fuzzy) | | | Importance | Performance |
|---|---|---|---|---|---|---|---|---|
| | *l* | *m* | *u* | *l* | *m* | *u* | | |
| IS1 | 0.00234 | 0.01061 | 0.03298 | 0.00371 | 0.028723 | 0.10085 | 0.01531 | 0.044428 |
| IS2 | 0.00699 | 0.030935 | 0.13888 | 0.01664 | 0.152217 | 0.47886 | 0.058935 | 0.215906 |
| IS3 | 0.00786 | 0.03043 | 0.10292 | 0.02523 | 0.09966 | 0.20114 | 0.04707 | 0.108677 |
| IC1 | 0.01256 | 0.0305 | 0.07627 | 0.01674 | 0.120447 | 0.31218 | 0.039777 | 0.149789 |
| IC2 | 0.00926 | 0.036875 | 0.11918 | 0.02755 | 0.14461 | 0.47864 | 0.055105 | 0.216933 |
| IC3 | 0.01809 | 0.04019 | 0.09995 | 0.00426 | 0.078067 | 0.20155 | 0.052743 | 0.094626 |
| EO1 | 0.0071 | 0.04589 | 0.15208 | 0.0086 | 0.063163 | 0.18379 | 0.068357 | 0.085184 |
| EO2 | 0.00609 | 0.023985 | 0.06463 | 0.01116 | 0.073043 | 0.22443 | 0.031568 | 0.102878 |
| EO3 | 0.00206 | 0.013495 | 0.03281 | 0.00383 | 0.07472 | 0.30071 | 0.016122 | 0.12642 |
| EO4 | 0.00983 | 0.03455 | 0.09965 | 0.01072 | 0.083493 | 0.24285 | 0.04801 | 0.112354 |
| EO5 | 0.0075 | 0.046495 | 0.15205 | 0.00476 | 0.09848 | 0.37259 | 0.068682 | 0.15861 |
| EC1 | 0.00954 | 0.028145 | 0.06369 | 0.00958 | 0.07706 | 0.18161 | 0.033792 | 0.089417 |
| EC2 | 0.00604 | 0.02461 | 0.0953 | 0.00288 | 0.087913 | 0.31416 | 0.041983 | 0.134984 |
| EC3 | 0.01669 | 0.03105 | 0.0632 | 0.00623 | 0.115083 | 0.42014 | 0.03698 | 0.180484 |
| EC4 | 0.00349 | 0.04358 | 0.30544 | 0.00963 | 0.113313 | 0.31297 | 0.117503 | 0.145304 |

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
