# Peer review of "Sustainable and Circular Hotels and the Water–Food–Energy Nexus: Integration of Agrivoltaics, Hydropower, Solar Cells, Water Reservoirs, and Green Roofs"

_sustainability, doi:10.3390/su16051985_

Round 1

Reviewer 1 Report

Comments and Suggestions for Authors

The article is positively valued for publication. The subject of the topic is of great interest to the tourism sector and in the current context of global change, with an analysis of the water-food-energy-correct link from the holistic approach.

The article is well structured and widely developed in all chapters. The introduction, literature review, and Materials and methods are correct.

As the only observations of improvement:

In chapter 4, section 4.1 would include a location map of Ramsar within Iran.

In chapter 5, it is necessary to relate this dialogue with the theoretical framework through the incorporation of some literature references.

Reviewer 2 Report

Comments and Suggestions for Authors

From what sources is sustainability relevant to a hotel's 4- or 5-star rating? If such a rating is provided, the source of that information should be clearly stated.

Conversely, if such a rating is clearly stated, why not conduct this research by that rating?

The original paper should also clearly state in the first section what methods were used in the original paper, how the methods used in this paper were used in other papers, and what conclusions were derived from the methods.

For example, Fuzzy methods are not mentioned in the Introduction or Literature review but are suddenly explained in the Methods section.

The paper is structured so that the reader is forced to read the section without any explanation of Fuzzy and without knowing why the paper takes the approach that it does.

How does it compare to other hotels or architecture in Iran or in other countries?

Are the methods you have taken reproducible? Or has anyone who has seen this paper checked it from the viewpoint of how it could be applied to (their) own company?

Comments on the Quality of English Language

While the Introduction and Literature review sections in this paper are long, it is not clear which research is strongly related to this study.

This makes it difficult to understand the importance of this study.

Check the citation of the figure or caption. For example, Table 5 does not exist in this paper. For example, could the caption for Figure 4 be shorter? Or can it be divided into two figures?

Reviewer 3 Report

Comments and Suggestions for Authors

I recommend publishing this article in this form. The article brings a high degree of novelty in the field of the hospitality industry aiming at objectives of maximum importance for the development in the sustainable perspective of the global economy. I congratulate the authors for the research carried out with the hope that it can represent a starting point for practitioners.

Reviewer 4 Report

Comments and Suggestions for Authors

I thank the authors for reviewing their manuscript. Although the research question and topic is relevant, the study was not presented in a rigorous manner.

The title shows challenges and strategies but not opportunities, why?

An introduction section encompassing the summary of the current theoretical background, identified research gaps, study objectives, and applied methodology is missing. In the current version, sections 1.1; 1.2; 1.3 can be considered part of the literature review. These sections do not present your study but rather a general background of your research.

From the first sections, it is not clear what the objective of the study is and how you intend to carry out the research investigations. Section “3. Materials and methods” is not transparent and do not clarify why it is important to investigate these research topics with specific techniques such as fuzzy logic, Z-SWARA and Z-MARCOS. Most importantly, Z-SWARA methodology is mentioned but not clarified and applied.

More clarification needs to be given to the methodology, results, and theoretical and managerial implications. It is important to mainly highlight the research problem and show how these issues were investigated by providing valuable implications. The current version of the manuscript primarily emphasizes the implementation of the methodology rather than delving into the interpretations of the results.

There are errors in punctuation (i.e. line 251) and definition of acronyms (i.e. SOAR, SOPA, WASPAS). In addition, some sentences are redundant and repetitive. For example, in line 35 there is this sentence, "Sustainable development in the hospitality industry...," it is carried over later in line 40. In section 1.1 many concepts are repeated and redundant. This makes the reading discontinuous and not very fluent. Proofreading by a native English speaker is necessary.

Comments on the Quality of English Language

Moderate editing of English language required

Reviewer 5 Report

Comments and Suggestions for Authors

I appreciated that the authors’ efforted to study an exciting paper. However, the way to develop theories is weak for this study, particularly in the literature review section.

1. In the Introduction section, the authors should focus on the research gaps and the research objectives. Besides, some contents could be moved to the literature review section.

2. Regarding methodology. It is important to describe the SCOC Analysis in this manuscript and provide corresponding references to prove the importance of the method.

3. In this study, the section on practical and theoretical contributions must be systematized and more clearly written. There is a lack of comparison with existing studies to claim the authors' new findings.

Comments on the Quality of English Language

Extensive editing of English language and style required.

Round 2

Reviewer 4 Report

Comments and Suggestions for Authors

I thank the authors for conducting a considerable revision effort. All my concerns have been resolved. Therefore, the new version of the manuscript is ready for publication.

Comments on the Quality of English Language

Minor editing of English language required